# Real Life Clinical Impact of Antimicrobial Stewardship Actions on the Blood Culture Workflow from a Microbiology Laboratory

**DOI:** 10.3390/antibiotics10121511

**Published:** 2021-12-09

**Authors:** Jose Maria López-Pintor, Javier Sánchez-López, Carolina Navarro-San Francisco, Ana Maria Sánchez-Díaz, Elena Loza, Rafael Cantón

**Affiliations:** 1Servicio de Microbiología, Hospital Universitario Ramón y Cajal and Instituto Ramón y Cajal de Investigación Sanitaria (IRYCIS), 28034 Madrid, Spain; jmlph@hotmail.com (J.M.L.-P.); jsl1991@hotmail.es (J.S.-L.); anit.sdg@gmail.com (A.M.S.-D.); elena.loza@salud.madrid.org (E.L.); rafael.canton@salud.madrid.org (R.C.); 2Red Española de Investigación en Patología Infecciosa (REIPI), Instituto de Salud Carlos III, 28029 Madrid, Spain; 3CIBER de Enfermedades Infecciosas (CIBERINFEC), Instituto de Salud Carlos III, 28029 Madrid, Spain

**Keywords:** bacteremia, antimicrobial susceptibility testing (AST), accelerate, bloodstream infection, sepsis

## Abstract

Background: Accelerating the diagnosis of bacteremia is one of the biggest challenges in clinical microbiology departments. The fast establishment of a correct treatment is determinant on bacteremic patients’ outcomes. Our objective was to evaluate the impact of antimicrobial therapy and clinical outcomes of a rapid blood culture workflow protocol in positive blood cultures with Gram-negative bacilli (GNB). Methods: A quasi-experimental before–after study was performed with two groups: (i) control group (conventional work-protocol) and (ii) intervention group (rapid workflow-protocol: rapid identification by Matrix-Assisted Laser Desorption/Ionization-Time-Of-Flight (MALDI-TOF) and antimicrobial susceptibility testing (AST) from bacterial pellet without overnight incubation). Patients were divided into different categories according to the type of intervention over treatment. Outcomes were compared between both groups. Results: A total of 313 patients with GNB-bacteremia were included: 125 patients in the control group and 188 in the intervention. The time from positive blood culture to intervention on antibiotic treatment decreased from 2.0 days in the control group to 1.0 in the intervention group (*p* < 0.001). On the maintenance of correct empirical treatment, the control group reported 2.0 median days until the clinical decision, while in the intervention group was 1.0 (*p* < 0.001). In the case of treatment de-escalation, a significant difference between both groups (4.0 vs. 2.0, *p* < 0.001) was found. A decreasing trend on the change from inappropriate treatments to appropriate ones was observed: 3.5 vs. 1.5; *p* = 0.12. No significant differences were found between both groups on 7-days mortality or on readmissions in the first 30-days. Conclusions: Routine implementation of a rapid workflow protocol anticipates the report of antimicrobial susceptibility testing results in patients with GNB-bacteremia, decreasing the time to effective and optimal antibiotic therapy.

## 1. Introduction

One of the priorities of microbiology laboratories’ workflow is the quick diagnosis of sepsis. Bloodstream infections (BSI) are the main cause of morbidity and mortality in hospitalized patients and the timely start of appropriate antibiotic therapy has an important impact on patient’s outcomes [1]. Currently, the development of sensitive and specific techniques that report results in a shorter time is known as one of the fields of further investigation, and several studies have demonstrated the impact of these actions in the patient outcomes, and also in the cost effectiveness aspect [2]. Diverse methods aimed at getting earlier antimicrobial susceptibility testing (AST) results are under development, but only a few have the Food and Drugs Administration (FDA) authorization or the European Conformity (CE) marker. The European Committee on Antimicrobial Susceptibility Testing (EUCAST) have published clinical breakpoints for early reading of disk-diffusion antibiograms performed directly from positive blood cultures, and the Clinical and Laboratory Standards Institute (CLSI) has established preliminary criteria to read the results after 16–18 h of incubation [3,4]. Other methods, such as the Accelerate Pheno™ System that uses fluorescence in situ hybridization using peptide nucleic acid probes (PNA-FISH) technology based on hybridization, provides AST results in 7–8 h [5] and more recently the QMAC-dRAST (Quantamatrix, Inc., Seoul, Korea) which is an automated system based on microscopic detection of growth of bacteria embedded in an agarose gel and can perform AST directly from positive blood cultures in 6 h [6]. Molecular techniques are rapid and sensitive, but to date, they are not affordable for all laboratories [7]. There are also groups working on the use of several techniques, such as flow cytometry, droplet-based microfluidics, calorimetry, or spectroscopy, in order to obtain AST results in less than 2 h [8,9,10,11,12,13].

In addition, early AST results availability has a great impact on antibiotic treatment adjustment, outcome, and survival of patients, especially if they are critically ill, as well as on the institutional success of the antimicrobial stewardship programs (ASP) [14,15,16].

In our center, a new work procedure was validated for positive blood cultures with Gram-negative bacilli (GNB), which allows immediate MALDI-TOF identification and obtaining AST with a commercial microdilution system in less than 24 h using the same pellet used for MALDI-TOF identification [17]. The new method differs from the standard in the lack of necessity of waiting for 24 h for the bacterial growth. This protocol is simple, cheap, and accessible to all microbiology laboratories that dispose of MALDI-TOF identification and an automatic AST system. The main objective of our work is to assess the real clinical impact of the implementation of this procedure, as part of the antimicrobial stewardship activities carried out in our laboratory.

## 2. Methods

### 2.1. Study Design

The study was carried out from September 2016 to August 2017 at Ramón y Cajal University Hospital (Madrid, Spain), a tertiary hospital with 1161 beds and more than 30,000 admissions per year, in which all medical and surgical specialties are represented. At the time of performing the study, the Microbiology Department provided routine blood culture results from 8:00 am to 18:00 pm (Monday to Friday) and from 8:00 am to 15:00 pm (Saturday and Sunday).

The objective of this work was to evaluate the clinical impact of the routine implementation of a rapid blood culture workflow protocol in all blood cultures harboring GNB. This protocol was previously described for *Enterobacterales* and *Pseudomonas aeruginosa* isolates [17]. Briefly, positive blood culture bottles were processed to rapidly obtain a bacterial pellet and streaked on solid media blood agar (BA) and chocolate agar (ChA). This pellet was used for MALDI-TOF identification and to inoculate the semiautomatic AST panels. A suspension was prepared directly from the pellet using the Prompt Inoculation System Wands. This AST technique agreed with standard evaluation criteria (<10% of total errors, including <1.5% VME [Very Major Errors], and <3% ME [Major Errors]; and at least 90% agreement in AST results). To evaluate the clinical impact of this protocol on patients’ antimicrobial treatment and outcome, a quasi-experimental before and after study was performed. Two groups of patients were established in the study: (i) the control group (conventional work protocol, used until December-2016) and (ii) the intervention group (rapid workflow protocol, used from January 2017). Figure 1 summarizes the workflow of both protocols.

In order to analyze the type of intervention over antimicrobial treatment, patients from both groups were classified into five different categories: (1) implementation of correct antibiotic treatment in patients without previous one; (2) maintenance of correct empirical treatment; (3) change to correct antibiotic treatment when empirical one was inappropriate; (4) de-escalation from broad-spectrum antibiotic according to clinical guidelines to another one more adequate based on AST report; and (5) other options. The antibiotic treatment was considered correct if the isolated microorganism was susceptible to it according to EUCAST breakpoint criteria, version 9.0 (https://www.eucast.org/fileadmin/src/media/PDFs/EUCAST_files/Breakpoint_tables/v_9.0_Breakpoint_Tables.pdf, accessed on 1 December 2021) to the antibiotic evaluated. The route of administration and the posology of antibiotics were not evaluated.

All positive blood cultures with GNB processed during the defined study periods were included. Patients with polymicrobial blood cultures, those with Gram-positive, anaerobic microorganisms or yeast, those in which the patient was receiving five or more antibiotics at the time of blood culture was obtained and patients with second or successive positive blood cultures with the same microorganism during admission, if the modification of the antibiotic treatment was carried out five or more days later than the final AST was available or when none of the interventions contemplated was carried out, were excluded.

The study was approved by the local Ethics Committee as it is stated in the record number 332 of this committee.

### 2.2. Data Collection

Clinical data including demographic information, suspected source of bacteremia, length of stay (LOS), admission ward, and previous baseline characteristics of patients from both groups were collected from clinical records.

The time from the positivity of the blood culture to the definitive AST report, the time from the positive blood culture to the intervention on antibiotic treatment, mortality at day-7 and day-30 from bacteremia, readmissions during the first 30 days after bacteremia, and admission to the intensive care unit (ICU) after bacteremia were analyzed. Antibiotic treatment information was collected from the electronic prescribing program of the Pharmacy Department.

### 2.3. Statistical Analysis

Statistical analysis was performed by STATA (13.0 Version). Results for continuous variables are expressed as means with standard deviation (SD) or as medians with first and third quartiles, while data for categorical variables are expressed as percentages. Comparative results between both groups were obtained through the Student’s *t*-test (normal distribution) or the Mann–Whitney U test (not normal distribution) for continuous variables and by chi-square analysis for categorical variables. The Shapiro–Wilk test was utilized to determine the kind of distribution. A *p*-value < 0.05 was considered statistically significant.

## 3. Results

One hundred and twenty-five patients were included in the control group and one hundred and eighty-eight in the intervention group. Baseline characteristics of patients included in each group are described in Table 1. No significant differences were found between both groups except in the type of microorganisms isolated, being more frequent *Escherichia coli* in the control group than in the intervention group (*p* = 0.023).

Time from the positivity of the blood culture to intervention on antibiotic treatment decreased from two days in the control group to one day in the intervention group (*p* < 0.001) (Figure 2).

Table 2 shows the time (expressed as median days) from positivity to antibiotic treatment intervention according to the category of intervention and ward of admission. Time to intervention decreased from two days in the control group to one day in the intervention group (*p* < 0.001). The decrease was significative in category 2 (maintenance of empiric treatment) from two days to one day; *p* < 0.001 and in category 4 (de-escalation) from four days to two days; *p* < 0.001. These differences could be observed both in medical and surgical wards.

There were no differences on LOS between both groups: 8.00 (8.00–16.00) vs. 8.00 (5.00–16.00); *p* = 0.756. There were also no significant differences based on the admission ward. In medical departments, LOS decreased from 9.00 (5.00–18.00) to 8.00 (5.00–15.00), *p* = 0.246; on surgical ward it changed from 9.00 (5.00–15.00) to 9.50 (5.00–15.00), *p* = 0.923; on infectious diseases ward, from 5.00 (3.00–10.00) to 6.00 (5.00–9.00), *p* = 0.318; and on ICU from 9.00 (7.00–16.00) to 23.00 (7.00–34.50), *p* = 0.109 (Appendix A).

Regarding outcomes, no significant differences were found between both groups: 7-days mortality ranged from 4% (5/125) to 6% (12/188), *p* = 0.362; 30-days mortality changed from 5% (6/125) to 8% (15/188), *p* = 0.271; and readmissions in the first 30 days after bacteremia episode varied from 10% (13/125) to 15% (29/188), *p* = 0.201 (see specific data on Appendix A). In medical wards when a de-escalation of antibiotic treatment (category 4) was performed, the rate of readmissions at 30 days was significantly higher in the intervention group (8% vs. 35%, *p* = 0.019).

## 4. Discussion

The implementation of this new protocol for processing positive blood cultures with *Enterobacterales* or *P. aeruginosa* advance AST results by one day, which impacts the antibiotic management of these bacteremic patients. The importance of a rapid adjustment to the correct antibiotic therapy is widely recognized [14,15], not only in what refers to the adequate coverage of the microorganism involved but also in the adjustment of the antibiotic spectrum to avoid the emergence of multidrug-resistant microorganisms or infections by *Clostridioides difficile* [18].

Microbiology laboratories can actively develop antimicrobial stewardship activities aimed at reducing the time for microorganisms identification and AST reporting and consequently helping to decrease the length of broad-spectrum antibiotics administration [15]. In addition to the ecological impact, the adjustment of empirical antibiotic therapy plays an important role in patient safety, since it allows to reduce potential toxicities and hypersensitivity reactions [19].

The clinical impact of reducing the time of AST obtaining varies depending on the intervention performed on antibiotic treatment after this information. However, multiple conditions such as the severity of the patient, adequacy of empirical treatment both in terms of susceptibility of the microorganism and in terms of pharmacokinetic/pharmacodynamic (PK/PD) and route of administration, source of bacteremia or focus control can also have an influence. This study has only evaluated the impact on the modification of the treatment, which has limited its results.

Our study did not find statistically significant differences in categories 1 (implementation of correct antibiotic), 3 (changing from an incorrect empirical treatment to a correct one), and 5 (other situations) in terms of mortality, readmissions, or early adjustment of antibiotic treatment, in concordance with other studies [15]. This is potentially due to the small number of patients included in each of these categories.

The decision to maintain an empirical antibiotic treatment (category 2) should always be based on the final AST report so that the empirical treatment becomes a targeted treatment. The data observed in our study in this category is of paramount importance due to several reasons. First, it demonstrates the high adherence of prescribers to the clinical guidelines confirmed by the vast majority of the correct empirical treatments, and second, the fact of initiating a correct empirical treatment is known to increase survival and decrease morbidity and mortality in patients with bacteraemia [20,21,22]. For this reason, it is of utmost importance the correct prescription and the rapid administration of the right empirical antibiotic treatments. However, in this intervention, advancing the result of the AST does not have an impact in terms of modification of the prescribed treatment but allows us to confirm the correctness of it earlier.

On the other hand, category 4 comprises patients in whom after a correct empirical treatment the definitive one is adjusted based on the final AST. In these patients, an earlier de-escalation allows a decrease in antibiotic pressure within our hospital, and secondarily it may help in reducing the emergence of antibiotic resistance and the prevalence of infections related to high antibiotic pressure such as *C. difficile* diarrhoea. However, in the data of our series, at least in the subgroup of patients admitted to medical wards, the fact of de-scaling the antibiotic treatment earlier, has been significantly related to an increase in readmissions in the first 30 days contrary to what is described by other authors [23]. This leads us to think that it is necessary not only to evaluate the information provided by the AST but also that the patient should be a focus globally, considering the probable source of bacteraemia, the clinical evolution with the prescribed empirical treatment, and the severity of the patient.

Unlike other studies, in which a multidisciplinary team performed interventions over a selected group of patients with the objective of early detection of sepsis and triggering the activation of the necessary measures to improve their prognosis (Survival Sepsis Campaign) [24], it should be noted that this improvement action was carried out only from the microbiology laboratory. During the study, we did not have available in our center a proper multidisciplinary stewardship program that intervened in bacteraemia, nor did we have “bundles” to improve the management of bacteraemia and the data were collected from all GNB bacteraemia and not only from those that met sepsis criteria. This is probably one of the reasons why the clinical impact observed after the intervention was limited.

Interestingly, patients from the intervention group experimented with more readmissions in the first 30 days after the bacteremia episode. A possible explanation could be the existence of infections produced by non-cultivable microorganisms sensitive to empirical treatments. The early suspension of empirical wide-spectrum antibiotics could permit the re-grow of these microorganisms and the development of critical conditions. Even, the increment of mortality of ICU patients in the intervention group could be related to this phenomenon. However, it is difficult to demonstrate, due to the limited number of patients included. This finding may also be due to the multiple statistical tests that were performed with various objectives in multiple categories, with the possibility that this result is statistically significant only by coincidence.

One of the fundamental limitations of our study is the collection of data in days and not in hours, due to the type of computer system used. This is an important limitation, since in these types of situations every hour counts and can impact on patient’s prognosis.

On the other hand, the main limitation of this work is that no data on the route of administration or dosage of the antibiotic treatment have been collected, which reduces the evaluation of the intervention only to the microbiological interpretation. Therefore, problems that may have existed in relation to the dose (adjustment of weight and/or renal function), the route of administration (oral or intravenous), and other PK/PD circumstances are not evaluated. This would be a useful approach to take into account when performing a similar study using the new EUCAST category “I” (susceptible, increased exposure) definition [25]. Another limitation of our work is that the severity of the infection (e.g., Pitt Score, Sepsis grade…) has not been considered to stratified patients; only being reflected if the patients were admitted in the ICU. With our data, we cannot affirm that the improvement in the time of obtaining the AST report reduces the number of admissions to the ICU. This may be explained because in bacteraemia, admission to the ICU usually occurs within the first 24–48 h after the beginning of the episode, which is the minimum time needed to obtain AST results using this method. Other parameters to evaluate the impact of advancing AST results in bacteraemia is the consumption of broad-spectrum antibiotics. Generally, this is associated with a decrease in DDD (defined daily dose)/100 stays and secondarily they achieve a reduction in cost [26]. The design of a study focused on the evaluation of the economic impact is required to confirm this hypothesis.

As already mentioned, our study has only evaluated the clinical impact of advancing the outcome of the AST on the modifications made in the antibiotic treatment. The study was carried out at a time when in our center there did not have an active antimicrobial stewardship program that allowed directing actions aimed at improving the attention to bacteraemia. Therefore, we believe that all those improvement activities that start from a microbiology laboratory must be accompanied by other measures (“bundle”) or be part of multidisciplinary activities that guarantee that the effort to obtain fast and reliable results is followed by measures, which applied to the patient, impact on their mortality, survival, and readmissions.

## 5. Conclusions

We have observed that rapid identification and AST of GNB in bacteremic patients leads to an earlier adjustment in antibiotic therapy and, therefore, to a narrower spectrum that provides adequate coverage. Rapid bacterial diagnostic procedures are strongly recommended to be implemented in the clinical microbiology laboratory. However, due to a high adherence of clinical practitioners to the clinical guidelines along with a broad prescription of correct empirical antibiotic therapy, these actions do not lead to a reduction in mortality.

## Figures and Tables

**Figure 1 antibiotics-10-01511-f001:**
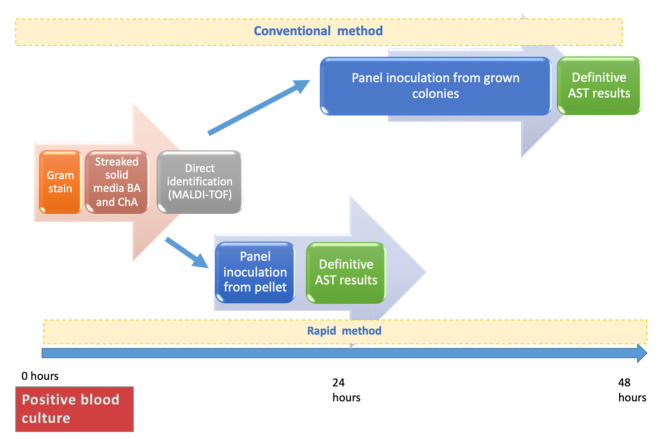
Temporal comparison between conventional and rapid working methods of positive blood cultures.BA: blood agar. ChA: chocolate agar. AST: antimicrobial susceptibility testing.

**Figure 2 antibiotics-10-01511-f002:**
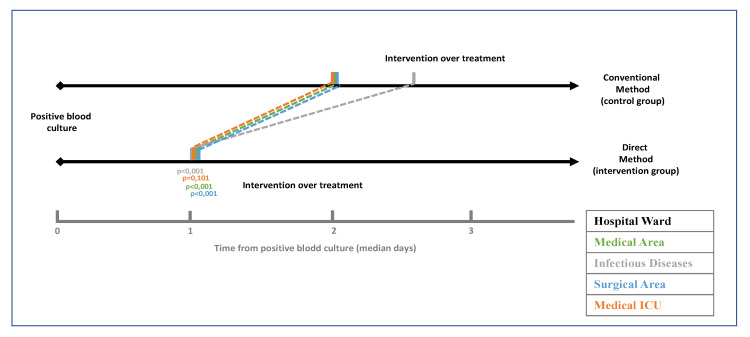
Temporary gain between control group and intervention group broken down by admission wards.

**Table 1 antibiotics-10-01511-t001:** Baseline characteristics of patients included in control and intervention groups.

	Characteristics	Control Group (*n* = 125)	Intervention Group (*n* = 188)	*p*
Comorbidities	Median Age	69.8 (66.5–73.1)	72.7 (70.2–75.1)	0.906
	Gender male	65 (52.0%)	109 (58.0%)	0.324
	Diabetes mellitus	31 (24.8%)	42 (22.3%)	0.645
	Chronic renal insufficiency	29 (23.4%)	58 (30.8%)	0.153
	Chronic pulmonary disease	20 (16.0%)	40 (21.3%)	0.258
	Heart failure and cardiovascular diseases	80 (64.0%)	124 (66.0%)	0.720
	Immunosuppression	24 (19.2%)	28 (14.9%)	0.324
	Neoplasia	44 (35.2%)	73 (38.8%)	0.524
	Neutropenia	7 (5.6%)	10 (5.3%)	0.919
	Organ transplant	7 (5.6%)	4 (2.1%)	0.104
Suspected source of bacteraemia	Urinary	74 (59.2%)	102 (54.3%)	0.393
	Abdominal	33 (26.4%)	45 (23.9%)	0.623
	Respiratory	10 (8.0%)	20 (10.6%)	0.447
	Catheter	4 (3.2%)	11 (5.9%)	0.420
	Skin and soft tissue	3 (2.4%)	7 (3.7%)	0.755
	Other	1 (0.8%)	3 (1.6%)	0.923
Microorganisms isolated	*E. coli*	87 (69.6%)	107 (56.9%)	0.023
	ESBL *E. coli*	10 (8.0%)	14 (7.5%)	0.866
	*K. pneumoniae*	18 (14.4%)	25 (13.3%)	0.785
	ESBL *K. pneumoniae*	0 (0%)	4 (2.1%)	0.262
	*P. aeruginosa*	2 (1.6%)	4 (2.1%)	0.936
	Others	8 (6.4%)	34 (18.1%)	0.003

ESBL: extended spectrum beta-lactamase; Bold letter: statistically significant.

**Table 2 antibiotics-10-01511-t002:** Time (median days (IQ1–IQ3)) from positive blood culture until intervention on the antibiotic treatment.

Global	Control Group	*n*	Intervention Group	*n*	*p*
General	2.00 [2.00–3.00]	125	1.00 [1.00–2.00]	188	**<0.001**
Category 1	4.00 [4.00–4.00]	1	2.00 [1.00–3.00]	12	0.166
Category 2	2.00 [2.00–2.00]	79	1.00 [1.00–1.00]	120	**<0.001**
Category 3	3.50 [3.00–4.00]	2	1.50 [1.00–2.00]	10	0.092
Category 4	4.00 [3.00–5.00]	36	2.50 [2.00–3.00]	34	**<0.001**
Category 5	2.00 [2.00–3.00]	7	1.50 [1.00–3.00]	12	0.329
Medical ward	2.00 [2.00–3.00]	84	1.00 [1.00–2.00]	121	**<0.001**
Category 1	--	0	1.50 [1.00–3.00]	6	--
Category 2	2.00 [2.00–2.00]	51	1.00 [1.00–1.00]	78	**<0.001**
Category 3	3.50 [3.00–4.00]	2	2.00 [1.00–2.00]	5	0.073
Category 4	3.00 [3.00–5.00]	26	3.00 [2.00–3.00]	23	**0.015**
Category 5	2.00 [2.00–3.00]	5	2.00 [1.00–3.00]	9	0.445
Surgical ward	2.00 [2.00–2.00]	18	1.00 [1.00–2.00]	38	**<0.001**
Category 1	--	0	2.00 [2.00–8.00]	3	--
Category 2	2.00 [2.00–2.00]	12	1.00 [1.00–1.00]	23	**<0.001**
Category 3	--	0	5.00 [2.00–8.00]	2	--
Category 4	4.00 [3.00–4.00]	5	2.00 [1.50–3.50]	8	**0.05**
Category 5	2.00 [2.00–2.00]	1	2.00 [1.00–3.00]	2	1
Infectious diseases	2.50 [2.00–3.00]	14	1.00 [1.00–1.00]	17	**<0.001**
Category 1	4.00 [4.00–4.00]	1	2.00 [1.00–3.00]	2	0.221
Category 2	2.00 [2.00–3.00]	11	1.00 [1.00–1.00]	14	**<0.001**
Category 3	--	0	--	0	--
Category 4	5.50 [4.00–7.00]	2	--	0	--
Category 5	--	0	1.00 [1.00–1.00]	1	--
ICU	2.00 [2.00–3.00]	9	1.00 [1.00–1.50]	12	0.101
Category 1	--	0	2.00 [2.00–2.00]	1	--
Category 2	2.00 [2.00–2.00]	5	1.00 [1.00–1.00]	5	**<0.001**
Category 3	--	0	1.00 [1.00–1.00]	3	--
Category 4	4.00 [3.00–5.00]	3	1.00 [1.00–5.00]	3	--
Category 5	2.00 [2.00–2.00]	1	--	0	--

(1) implementation of correct antibiotic treatment in patients without previous one; (2) maintenance of correct empirical treatment; (3) change to correct antibiotic treatment when empirical one was inappropriate; (4) de-escalation from a correct empiric antibiotic treatment and (5) other options; Bold letter: statistically significant.

## Data Availability

Data are available on request to the investigators.

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
