# Peer review of "Real Life Clinical Impact of Antimicrobial Stewardship Actions on the Blood Culture Workflow from a Microbiology Laboratory"

_antibiotics, 2021, doi:10.3390/antibiotics10121511_

Round 1

Reviewer 1 Report

The authors compare the conventional work protocol versus an experimental workflow according to the type of intervention over treatment and the outcomes were compared between both groups to assess the impact of antibiotic stewardship. The results have shown that rapid workflow protocol anticipates the report of antimicrobial susceptibility testing results in patients with GNB-bacteremia, decreasing time to effective and optimal antibiotic therapy. 
The abstract is well structures properly written and contains enough information. 
How ever, few things need to be sorted out.
What is the meaning of “On maintenance of a correct empirical treatment”?
Line 33: either quick or acute, not both, otherwise, it is not clear to me.
Line 36-37: “currently…. Further…” + “past participle...” , please rephrase this sentence.
Line 39: “and also in the economic aspect” = “cost effective”?? 
I am wondering why the authors wrote the manuscripts in past tense. 
Ex: “The main objective of our work was to assess the real clinical impact
of the implementation of this procedure, as part of the antimicrobial stewardship activities carried out in our laboratory.” 
this sentence, the objective should be put in present tense.
Line 71-72 I think the protocol you are evaluating should be properly described in this manuscript even if it has been described elsewhere. 
Please specify if the samples/indivuals were randomly selected for each group.
Why do you use blood agar  and chocolate agar to isolate gram negative strains since you already know that GN are growing in your blood culture? Please clarify.
I think usually figure's title are at the bottom and table’s on the top.
The reason why wenormaly  prepare a fresh suspension for the AST panel is to have bacteria at the growing stage 6-18h if I am correct and your experimental protocol is saying that this is no more necessary, correct?
Line 92: do you mean patient will be receiving 5 or more antibiotherapy at the same time??
statistical analysis 
line 112-116: “ Comparative results between both groups were obtained through Student ́s t test (normal distribution) or Mann-Whitney U test (not normal distribution) for continuous variables and
by chi-square analysis for categorical variables. A p-value <0.05 was considered statistically significant.”
can you please clarify in which case you have used the Student’s t-test? How do you know that both group are normally distributed? Age( mean=median?), what about Mann-Whitney U test? Chi-square?etc.. please bear in mind that the statistical significance of a test is only evaluating the robustness of the test not the actual biological interpretation of the data.
Table1: what is the sens of comparing basic characteristics of both groups without intervention using p-value? How are you going to interpret your p-value?
Also, it is difficult to assess the difference between 1 and 2 days, can you please use hours instead of days?
Your study design is critical and depend on your initial hypothesis. There is one important things that I was expecting to see is the AST concordance between the conventional method and the experimental method, to realize that your groups are composed of different individuals. How do we measure the impact of the AST from MALDITOF pellet on the resistance profile?

Author Response

Line by line reply to reviewers

Reviewer 1

Initial lines

Reviewer’s Comment

Lines in Revised Document

Response

The authors compare the conventional work protocol versus an experimental workflow according to the type of intervention over treatment and the outcomes were compared between both groups to assess the impact of antibiotic stewardship. The results have shown that rapid workflow protocol anticipates the report of antimicrobial susceptibility testing results in patients with GNB-bacteremia, decreasing time to effective and optimal antibiotic therapy. 

Thank you for complementing the study. We have followed your suggestions

Our main conclusion is that a rapid identification and AST of GNB in bacteremic patients have an impact on the adjustment in antibiotic therapy.

The abstract is well structures properly written and contains enough information. 

Thank you for your comments. They are very welcome.

What is the meaning of “On maintenance of a correct empirical treatment”?

The implementation of a correct empirical treatment is known to increase the survival and decrease morbidity and mortality. Also, it reflects the level of adherence of prescribers to the clinical guidelines.

But the adequacy of the empirical antibiotic treatment is not known until the result of the AST is available. Thus, although the empirical treatment could be effective, is important accelerate the result of AST to be sure t of the suitability of the chosen treatment.

33

Line 33: either quick or acute, not both, otherwise, it is not clear to me.

33

Done

36-37

Line 36-37: “currently…. Further…” + “past participle...” , please rephrase this sentence.

37

Done

39

Line 39: “and also in the economic aspect” = “cost effective”?? 

39

We have rephrase the sentence.

I am wondering why the authors wrote the manuscripts in past tense. 
Ex: “The main objective of our work was to assess the real clinical impact
of the implementation of this procedure, as part of the antimicrobial stewardship activities carried out in our laboratory.” 
this sentence, the objective should be put in present tense.

60

Done

71-72

Line 71-72 I think the protocol you are evaluating should be properly described in this manuscript even if it has been described elsewhere. 

73-78

We have included some comments in lines 73-78.

Please specify if the samples/indivuals were randomly selected for each group.

81

Done

Why do you use blood agar and chocolate agar to isolate gram negative strains since you already know that GN are growing in your blood culture? Please clarify.

The usual laboratory routine was combined at the same time with the implementation of new measures during the study. A mention is included in the text.

I think usually figure's title are at the bottom and table’s on the top.

86 and 136

We have corrected it.

The reason why we normally prepare a fresh suspension for the AST panel is to have bacteria at the growing stage 6-18h if I am correct and your experimental protocol is saying that this is no more necessary, correct?

Results should be interpreted with caution, but our study affirms that pellet obtained the same day of the positivity of the blood culture would be enough in GNB.

The bacterial recount is critical to obtain an acute result of AST, but in a previous work we studied and estimated the bacterial count precedent of pellet in GNB (ref. 17)

92

Line 92: do you mean patient will be receiving 5 or more antibiotherapy at the same time??

Yes, patients with a therapy that included 5 or more antibiotics at the time of obtain blood cultures were not included in the study. This profile of patients was more frequent in ICU. But in this study only two patients was excluded for this reason.

112-116

line 112-116: “Comparative results between both groups were obtained through Student ́s t test (normal distribution) or Mann-Whitney U test (not normal distribution) for continuous variables and by chi-square analysis for categorical variables. A p-value <0.05 was considered statistically significant.”
can you please clarify in which case you have used the Student’s t-test? How do you know that both groups are normally distributed? Age(mean=median?), what about Mann-Whitney U test? Chi-square?etc.. please bear in mind that the statistical significance of a test is only evaluating the robustness of the test not the actual biological interpretation of the data.

122-123

We utilized the Shapiro-Wilk test to determine the normal (or not) distribution. Student´s t-test was finally applied only to age of patients, and the rest of parameters were compared using Mann-Whitney U test.

Finally, we used Chi-square test to compare between categorical variables. Mainly, we used it to compare baseline characteristics of patients.

Table1: what is the sense of comparing basic characteristics of both groups without intervention using p-value? How are you going to interpret your p-value?

These data are presented to verify that there were no significant differences in the patients included in both periods and that therefore there were no significant differences in terms of age, sex, baseline situation and comorbidities ... which could interfere as a confounding factor in the interpretation of the results.

Also, it is difficult to assess the difference between 1 and 2 days, can you please use hours instead of days?

We performed our study using days because because the work routine of the laboratory does not allow to have the time in exact hours.

Your study design is critical and depend on your initial hypothesis. There is one important thing that I was expecting to see is the AST concordance between the conventional method and the experimental method, to realize that your groups are composed of different individuals. How do we measure the impact of the AST from MALDITOF pellet on the resistance profile?

76-78

The validation of the rapid blood culture workflow protocol was performed in a previous work (ref. 17)

You can consult the details in:

López-Pintor, J.M.; Francisco, C.N.-S.; Sánchez-López, J.; García-Caballero, A.; de Bobadilla, E.L.F.; Morosini, M.I.; Cantón, R. Direct antimicrobial susceptibility testing from the blood culture pellet obtained for MALDI-TOF identification of Enterobacterales and Pseudomonas aeruginosa. Eur. J. Clin. Microbiol. Infect. Dis. 2019, doi:10.1007/s10096-019-03498-y

We have included a comment in lines 76-78:

“This AST technique is in agreement with standard evaluation criteria (< 10% of total errors, including < 1.5% VME, and < 3% ME; and at least 90% agreement in AST results)”

Reviewer 2 Report

This article by Lopez-Pintor et al., looks at the possible effects of a more rapid antibiotic sensitivity screening approach for Gram-negative bacteria. This is an interesting area and could help drive better use of antibiotics and patient care. However, the conclusions were not written in the clearest way, so that I was left wondering what the main conclusions really were. I think that if the discussion was made more clear and explicit then it would greatly improve this manuscript.

 In this paper I found numerous grammatical errors- I have pointed out a couple below, but there are several others that will need some careful editing. A few examples:

  1. Line 12: Accelerated not accelerate
  2. Use of . instead of , in numbers. Suggest change to 1,161 etc.
  3. Results: first paragraph- a lot of Chinese? Symbols 错误!未找到引用源
  4. Table 2: x axis: positive blodd culture- should be blood culture.
  5. Line 152-53 : “…fast antibiotic adjust is widely known..” recommend change to “..rapid adjustment to the correct antibiotic therapy is widely recognized..”

General comments:

Introduction:

Can the authors provide a more detailed overview of how the new rapid workflow differs from the standard workflow? Is it just the automation of what would normally be carried out by staff?

Methods:

Why were patients with a second bacteraemia excluded?

Results:

In table 1, sources of bactaeremia are listed, adding up to 100%. What about patients where no source could be identified? It would be unusual for all positive GNB blood cultures to have a matched culture from the source, or a clear source based on clinical signs.

In table 2, the relevance of the categories isn’t clear. Can the authors explain in the methods section what the categories represent? Or can the table be rewritten to have more descriptive subgroups instead of the word “category”? I notice that these are outlined in the discussion, which is not useful.

In table 2, the results look as though time to result was measured in whole days (or whole and half days) rather than hours? If hours were used, one would expect to see the quartiles as a range or different numerical values. Can the authors clarify? Does this have to do with the timepoints at which AST is measured in the rapid workflow (I.e. in quanta of days?)

One sentence is dedicated to table 2 in the results text. Could the authors expand on this to highlight relevant findings from table 2?

The authors mention that the aim of the study is to assess the real clinical impact of the intervention, however the data for this is contained in supplementary tables. Could this data be included in the main text? Was ICU LOS measured? This would be an interesting variable to assess.

Discussion:

Overall, I thought the conclusions were not well presented. Was the conclusion that the intervention was not worthwhile? Or was the conclusion that the intervention might be worthwhile but the number of patients was too low to give a statistically significant result? Was a power calculation done to work out the number of patients that would have to be involved to produce significant results?

One of the most interesting findings from this study is the increased readmissions in patients with earlier de-escalation of antibiotics. Could the authors also discuss the possibility that empirical therapy may be covering for non-culturable infections where the correct antibiotic would never be known. The authors could also consider pointing out that there could be a signal with higher mortality in the ICU patients with earlier de-escalation, but greater numbers of patients are required to answer this.

Author Response

Line by line reply to reviewers

Initial lines

Reviewer’s Comment

Lines in Revised Document

Response

This article by Lopez-Pintor et al., looks at the possible effects of a more rapid antibiotic sensitivity screening approach for Gram-negative bacteria. This is an interesting area and could help drive better use of antibiotics and patient care. However, the conclusions were not written in the clearest way, so that I was left wondering what the main conclusions really were. I think that if the discussion was made more clear and explicit then it would greatly improve this manuscript.

Thank you for complementing the study and writing quality.

We have followed your suggestions in order to improve the quality of the manuscript.

In this paper I found numerous grammatical errors- I have pointed out a couple below, but there are several others that will need some careful editing. A few examples:

Several changes along the manuscript

We have taken in account your tips. Also, the text has been revised by a native English speaker.

12

Line 12: Accelerated not accelerate

12

Corrected

Use of . instead of , in numbers. Suggest change to 1,161 etc.

Several changes along the manuscript

Done

Results: first paragraph- a lot of Chinese? Symbols 错误!未找到引用源

There is no Chinese symbols in our text, may be a problem with the archive?

Table 2: x axis: positive blodd culture- should be blood culture.

144

Done

152-153

Line 152-53: “…fast antibiotic adjust is widely known..” recommend change to “..rapid adjustment to the correct antibiotic therapy is widely recognized..”

165-166

Done

Introduction:

Can the authors provide a more detailed overview of how the new rapid workflow differs from the standard workflow? Is it just the automation of what would normally be carried out by staff?

57-58

We have added some comments about the difference between new and standard workflow.

99-100

Methods:

Why were patients with a second bacteraemia excluded?

Patients with second or successive bacteremic episodes during admission were excluded because normally patients receive antibiotic therapy in their first episode of bacteremia, so it would be difficult interpretate the adequation of the previous therapy and the change to another one in a hypothetical second episode.

Also, we avoid duplicate baseline characteristics of the same patient.

Table 1

Results:

In table 1, sources of bactaeremia are listed, adding up to 100%. What about patients where no source could be identified? It would be unusual for all positive GNB blood cultures to have a matched culture from the source, or a clear source based on clinical signs.

107 and Table 1

We have selected the source suspected by the clinician responsible of each patient.

We have changed “source of bacteraemia” by “suspected source of bacteraemia”

Table 2

In table 2, the relevance of the categories isn’t clear. Can the authors explain in the methods section what the categories represent? Or can the table be rewritten to have more descriptive subgroups instead of the word “category”? I notice that these are outlined in the discussion, which is not useful.

The mean of each category is explained in materials and methods (lines 89-92) and also in the legend of the table 2.

In table 2, the results look as though time to result was measured in whole days (or whole and half days) rather than hours? If hours were used, one would expect to see the quartiles as a range or different numerical values. Can the authors clarify? Does this have to do with the timepoints at which AST is measured in the rapid workflow (I.e. in quanta of days?)

We performed our study using days because because the work routine of the laboratory does not allow to have the time in exact hours.

One sentence is dedicated to table 2 in the results text. Could the authors expand on this to highlight relevant findings from table 2?

139-143

We have added some comments about table 2 in lines 139-143.

The authors mention that the aim of the study is to assess the real clinical impact of the intervention, however the data for this is contained in supplementary tables. Could this data be included in the main text? Was ICU LOS measured? This would be an interesting variable to assess.

149-154

The information is included in lines 149-154. ICU LOS was measured, but the result should be interpreted with caution due to the limit number of ICU patient included.

Outcomes were described in lines 155-161.

Discussion:

Overall, I thought the conclusions were not well presented. Was the conclusion that the intervention was not worthwhile? Or was the conclusion that the intervention might be worthwhile but the number of patients was too low to give a statistically significant result? Was a power calculation done to work out the number of patients that would have to be involved to produce significant results?

In the conclusions we mentioned that “rapid bacterial diagnostic procedures are strongly recommended to be implemented in the clinical microbiology laboratory”. However, and due to the limit number of patients included, the majority of the results were no significative. Anyway, the intervention could have a great impact in terms of cost-effectiveness, decrease of multi-resistance isolates, and stablish a correct antibiotic treatment.

A different thing is the impact on patient’s outcomes. Is difficult to impact on LOS or mortality only with a laboratory intervention, mainly because the high adherence of clinicians to the clinical guidelines of antibiotic empiric  treatments.

One of the most interesting findings from this study is the increased readmissions in patients with earlier de-escalation of antibiotics. Could the authors also discuss the possibility that empirical therapy may be covering for non-culturable infections where the correct antibiotic would never be known. The authors could also consider pointing out that there could be a signal with higher mortality in the ICU patients with earlier de-escalation, but greater numbers of patients are required to answer this.

221-227

These conclusions are difficult to demonstrate, but we consider of interest this point of view.

We have added some comments in the discussion section (lines 221-227)